# OpenReview forum: "ADMIT: Few-shot Knowledge Poisoning Attacks on RAG-based Fact Checking"
_ICLR.cc/2026/Conference — ICLR 2026 Conference Desk Rejected Submission_

### Official Review · Reviewer_jZ8Q · 2025-10-28

**Soundness:** 2
**Presentation:** 2
**Contribution:** 3
**Rating:** 4
**Confidence:** 3

**Summary:**

Authors propose a pipeline for attacking automatic fact checking systems in the setting that the attacker sees the system as a black box, and is allowed to  inject a small set of false statements in the retrieved evidence. Given a claim, a set of relevant evidence is obtained (either from the web, or synthetically), then an LLM is used to generate a piece of false evidence based on the obtained relevant evidence. The the claim is added as a prefix to the false evidence and inserted into the corpus to be retrieved by the fact checking system.

The pipeline is evaluated in four datasets, and shows substantial successful attack rates. The experiments are carried out over multiple LLMs (as fact checkers) and the results are consistent.

**Strengths:**

-  The experiments are extensive.
-  The performance in the proposed setting is very good.
-  The explanations are very detailed.

**Weaknesses:**

- Authors repeatedly claim that their setting is inspired by real world scenarios. But How realistic is it to have the exact claim beforehand? In their pipeline this is crucial as they use it to create the false evidence, and also to make it retrievable by pre-pending it to the false evidence. If it is not feasible, then what can be done? How would this change the experiments reported in the paper?
- Authors claim as if no study has previously done on advarsarial attacks on fact checking systems!! This is not true, there are many studies on this topic [1]. Authors will probably come up with justification and say that "well those studies have not looked at the problem from XXX aspect". The point is this will question the author's claim about the topic, and also reduces the study to just a simple pipline which relies on a "to be justifed" assumption.
- I appreciate the extensive experiments, but authors frequently refer to the appendix section inside the main body of the paper, which makes understanding the work (specifically for review purposes) very difficult.

[1] Adversarial Attacks Against Automated Fact-Checking: A Survey, 2025.

**Questions:**

None

---

> ### Author Response · Authors · 2025-11-30
> **Reponse to Reviewer jZ8Q (1/2)**
>
> **Q1**: Authors repeatedly claim that their setting is inspired by real world scenarios. But How realistic is it to have the exact claim beforehand? In their pipeline this is crucial as they use it to create the false evidence, and also to make it retrievable by pre-pending it to the false evidence. If it is not feasible, then what can be done? How would this change the experiments reported in the paper?
>
> ---
>
> **A1:** We believe there is a misunderstanding of our threat model and would like to clarify why knowing the target claim (or its factual locus) is both realistic and consistent with prior work.
>
> **First**, all existing RAG poisoning studies, including PoisonedRAG [1], RetrievalPoison [2], AgentPoison [3], GraphPoison [4] and related knowledge-poisoning work, assume that the attacker knows which fact or claim they aim to overturn. This is inherent to any targeted poisoning attack. Without knowing the target, an attacker would have no objective to optimize for. Importantly, the attacker does not need the “exact string” of the claim, merely the underlying fact that the system will be queried about.
>
> **Second**, our method does not rely on perfect retrieval. In our one-shot experiments, the LLM receives **four clean evidence passages and only one adversarial passage**, yet ADMIT still induces failures (e.g., an ASR of $0.44$ on FEVER with GPT-4o in Table 1). This shows that ADMIT **generalizes across different surface forms of the claim** and does **not** require a verbatim match (i.e., ADMIT nearly achieves perfect recall under the sparser retriever, Table 19). Recall that we append a rephrased query, rather than the original query, to the false evidence (Section 3.2).
>
> **Third**, in realistic settings the attacker can infer the target claim from publicly available text such as headlines, social media narratives, or ongoing public debates. What matters is the factual locus (e.g., “Did X happen?”), not the exact phrasing. ADMIT’s **observe-then-write** mechanism is explicitly designed to operate on such approximate or proxy formulations.
>
> In general, even without knowing the exact claim, ADMIT can **observe related facts on the web and craft tailored “fake’’ passages** that target any semantically related claims. Our results demonstrate this feasibility (Tables 19–22; Figures 4 and 13).
>
>
> [1] Zou, W.,et. al, 2025. PoisonedRAG: Knowledge corruption attacks to Retrieval-Augmented generation of large language models. In 34th USENIX Security 25 (pp. 3827-3844).
>
> [2] Zhong, Z., Huang, Z., Wettig, A. and Chen, D., 2023. Poisoning retrieval corpora by injecting adversarial passages. In EMNLP (2023).
>
> [3] Chen, Z., Xiang, Z., Xiao, C., Song, D. and Li, B., 2024. Agentpoison: Red-teaming llm agents via poisoning memory or knowledge bases. NIPS, 37, pp.130185-130213.
>
> [4] Liang, J., Wang, Y., Li, C., Zhu, R., Jiang, T., Gong, N. and Wang, T., 2025. Graphrag under fire. arXiv preprint arXiv:2501.14050.

---

> ### Author Response · Authors · 2025-11-30
> **Reponse to Reviewer jZ8Q (2/2)**
>
> **Q2**: Authors claim as if no study has previously done on advarsarial attacks on fact checking systems!! This is not true, there are many studies on this topic [5]. Authors will probably come up with justification and say that "well those studies have not looked at the problem from XXX aspect". The point is this will question the author's claim about the topic, and also reduces the study to just a simple pipline which relies on a "to be justifed" assumption.
>
>
> **A2:** We believe there is also a misunderstanding regarding the realistic scope of our work. Our target is **End2End LLM-based RAG fact checking in**, which is fundamentally different from the **previous** automated fact checking systems surveyed in [5].
> We carefully read through the survey paper [5], explain following difference:
>
> - (1) Most methods in survey [5] use BERT-style verifiers [6] or language model [7], whereas powerful LLMs—as verifiers—exhibit varying levels of robustness in debunking false evidence.
>
> - (2) Prior work in survey [5] typically examines adversarial attacks on **individual components** of a fact-checking system (i.e., directly evaluate the verifier ability to check facts like FACTEVAL [8]). While **we evaluate the entire RAG pipeline**, including a real retrieval corpus, simulated RAG behavior, and a clearly defined threat model.
>
> - (3) Although a few papers in [5] mention LLM-based attacks, **the survey explicitly cites PoisonedRAG as the only representative study on LLM-based System** in their section 7. Our work extends this emerging line of research rather than overlapping with attacks on pre-LLM fact checkers.
>
> Moreover, from a realistic perspective, fact-checking systems must provide persuasive and coherent explanations [9], an aspect overlooked in prior knowledge-poisoning work. ADMIT fills this gap by showing that an attacker can manipulate the entire RAG pipeline and induce persuasive explanations even under practical constraints.
>
>
> **Novelty of our threat model**.
>
>  Prior RAG poisoning attacks implicitly assume that the attacker can inject **many more poisoned passages than clean ones**, effectively dominating the retrieved context. This assumption is unrealistic: *real adversaries cannot overwhelm search engines, news archives, or public knowledge bases at scale*.
>
> Our **few-shot poisoning** setting captures a more realistic threat model, where only one or a few injected passages are permitted. This setting introduces **unique technical challenges** that cannot be solved by simply adapting prior pipelines.
>
> **Not a simple pipeline**.
>
>  Although our overall workflow appears modular, **few-shot poisoning is fundamentally harder** because the adversarial passage must simultaneously:
>  - (1) survive retrieval among primarily clean evidence,
>
> -  (2) override multiple clean documents during generation, and
>
>  - (3) remain statistically indistinguishable under defensive consolidation.
>
> Meeting all three constraints requires our observe-then-write strategy, decomposed query augmentation, and cross-model optimization. As shown in our ablations, naive adaptations of prior attacks fail under this stricter regime.
>
>
>
> [5] Adversarial Attacks Against Automated Fact-Checking: A Survey, 2025.
>
> [6] Devlin, J., Chang, M.W., Lee, K. and Toutanova, K., 2019, June. Bert: Pre-training of deep bidirectional transformers for language understanding. In NAACL (long and short papers) (pp. 4171-4186).
>
> [7] Thorne, J., Vlachos, A., Christodoulopoulos, C. and Mittal, A., 2019, November. Evaluating adversarial attacks against multiple fact verification systems. In EMNLP-IJCNLP (pp. 2944-2953).
>
> [8] Mamta, M. and Cocarascu, O., 2025, April. FactEval: Evaluating the Robustness of Fact Verification Systems in the Era of Large Language Models. In NAACl (Volume 1: Long Papers) (pp. 10647-10660).
>
> [9] Kotonya, N. and Toni, F., 2020. Explainable automated fact-checking: A survey. arXiv preprint arXiv:2011.03870.
>
> ---
>
> **Q3**: I appreciate the extensive experiments, but authors frequently refer to the appendix section inside the main body of the paper, which makes understanding the work (specifically for review purposes) very difficult.
>
> **A3**: We appreciate the reviewer’s feedback. We recognize that frequent appendix references may interrupt the reading flow. Because of strict page limits, we moved supplementary details, including extended tables, ablations, and qualitative analyses, to the appendix to keep the main text focused on core contributions.
>
> In the revision, we will improve readability by summarizing essential points directly in the main body and reducing unnecessary cross-references wherever possible.

---

### Official Review · Reviewer_8hnt · 2025-10-29

**Soundness:** 3
**Presentation:** 3
**Contribution:** 2
**Rating:** 4
**Confidence:** 4

**Summary:**

This paper studies knowledge poisoning attacks on RAG, where malicious documents are injected to manipulate outputs. The authors find that attack success decreases when correct documents are retrieved alongside the malicious ones. To address this, they propose simulating such scenarios by collecting or generating realistic correct documents, enabling the creation of stronger malicious ones. This approach improves over naive document generation methods.

**Strengths:**

1. The paper is well-written, clearly structured, and easy to follow.
2. The experiments are comprehensive and include comparisons with prior work.
3. The proposed method shows improvements over the naive document generation approach.

**Weaknesses:**

1. The contribution is incremental. Knowledge poisoning in RAG has been well studied, including improving retrieval and injecting malicious documents to manipulate model behavior. This work focuses on a specific detail and provides only a modest extension of existing ideas.
2. The novelty is limited. The core idea is simple—providing the LLM with correct documents and prompting it to generate improved malicious ones. Moreover, some components overlap with prior work, such as making malicious documents more retrievable by appending the target query, a strategy already used in PoisonedRAG (Zou et al.).
3. While the method achieves some improvement, the gains are modest (e.g., an average ASR increase of 8% in 1-shot). Given the paper’s focus on this specific enhancement, I expected more substantial improvements. In some cases, such as on Climate-FEVER (Table 1), the improvement is marginal.

**Questions:**

1. In the defense evaluation on LLM-based fake news detection, FakeWatch is tested only up to Llama-2-7B, which is not convincing evidence that LLM-based detection fails. Have the authors tried stronger models, such as GPT, for a quick check?
2. Why does the divide-and-vote defense fail? Does this imply that malicious documents dominate the retrieved results during the poisoning process?

---

> ### Author Response · Authors · 2025-11-30
> **Reponse to Reviewer 8hnt (1/3)**
>
> **W1: The contribution is incremental.**
>
> **A1:** We respectfully disagree that our contribution is merely incremental. Our work studies a qualitatively different and strictly **more constrained threat model** than prior RAG-poisoning attacks, and shows that this setting is both practically relevant and technically nontrivial.
> Most knowledge-poisoning work is based on the assumption that **editing public sources is feasible, as proposed in paper [1]**. However, existing methods often **overstate the attacker’s ability** by assuming they can inject many poisoned passages, which would likely be flagged as fake sources.
>
> For example, PoisonedRAG implicitly relies on malicious documents dominating the retrieved context by volume. In contrast, ADMIT targets a **few-shot poisoning** regime that better reflects realistic adversaries in fact-checking systems, where the goal is to flip a specific claim while inserting only a small number of documents within a large corpus.
>
> Concretely, we restrict the attacker to at most $m \leq k$ injected passages, where k denotes the number of retrieved documents. In fact-checking, this reflects a realistic setting in which the fact-checker can cross-reference credible sources rather than being driven by a small number of manipulated passages. Clean evidence remains in the retrieved context and directly competes with the poisoned passage. While prior work typically assumes $m \gg k$, so that the attacker can dominate the retrieval set.
> This leads to a different technical problem: ADMIT must optimize the *content* of a very small number of passages so that they both (i) **beat the clean evidence** in the model’s internal competition and (ii) **survive retrieval**, rather than relying on sheer quantity.
>
> **Our success criteria is also strictly stronger.**
>
> PoisonedRAG measures success **only by inducing an incorrect answer**. ADMIT requires **simultaneously flipping the model’s verdict and inducing a deceptive justification (high DJR)**, which is particularly important in fact checking settings where users rely on explanations, not just labels. Under these more restrictive constraints and under the same black-box access assumptions, ADMIT still achieves higher attack success rates. On average, we improve over PoisonedRAG by 11.2% ASR across four datasets (table 1), with gains up to 33% on FEVER (o1 mini) and 24% on SciFact, evaluated across 11 LLMs and 4 retrievers (table19-22), that is, at a scale considerably larger than prior work.
>
> Finally, we note that PoisonedRAG itself explicitly highlights fact checking as an open challenge, stating that poisoning in these settings and more resilient poisoned text are **important future directions**. Our work directly addresses this gap by (i) formalizing a more restricted and realistic threat model for fact checking RAG, (ii) designing content optimized poisoning strategies that remain effective when clean evidence is present, and (iii) evaluating transfer across a wide range of models and retrievers.
>
> We agree that we build on an existing line of knowledge poisoning work, but we believe that moving from high budget, context dominating poisoning to few-shot, fact checking oriented attacks that must coexist with clean evidence is a **substantive step rather than a minor variation**.
>
> [1] Carlini, N., Jagielski, M., Choquette-Choo, C.A., Paleka, D., Pearce, W., Anderson, H., Terzis, A., Thomas, K. and Tramèr, F., 2024, May. Poisoning web-scale training datasets is practical. In 2024 IEEE Symposium on Security and Privacy (SP) (pp. 407-425). IEEE.

---

> ### Author Response · Authors · 2025-11-30
> **Reponse to Reviewer 8hnt (2/3)**
>
> **W2**: The novelty is limited. Providing the LLM with correct documents and prompting it to generate improved malicious ones is simple. Moreover, some components overlap with prior work, such as making malicious documents more retrievable by appending the target query..
>
> **A2:** We respectfully disagree with the reviewer’s assessment on both fronts.
>
> First, the claim that “one can simply ask an LLM to generate improved malicious documents” overlooks a core difficulty in **black-box retriever & generator poisoning**. In realistic settings, **the attacker has no access to the factual evidence in knowledge base**, making it unclear what content should be mimicked, contradicted, or strategically distorted. Our attack therefore introduces an explicit **observe–then–write** paradigm: we reconstruct a *proxy view* of the victim’s observation space, then condition generation on this inferred evidence distribution. This is not a trivial prompting trick; it is a principled mechanism for poisoning under stricter budget constraints.
>
>
> Second, while prior work has experimented with query concatenation to increase retrievability, our method is **not** a simple reapplication of that idea. Direct concatenation is easily detectable by substring or overlap filters **as discussed in Sec 3.3**. ADMIT instead uses **rephrased/decomposed query** derived from the proxy environment, producing retrieval cues at a finer granularity that both (1) avoid naive detection heuristics and (2) significantly improve recall across dense retrievers (**from 82.4% to 95.7% in Table 3**). Importantly, ADMIT does **not** rely on injecting many poisoned text; even single malicious document injection suffices for the attack to succeed.
>
>
> Overall, our novelty lies not in isolated components, but in the coherent attack formulation that addresses more challenging black-box constraints while achieving strong, cross-model, few-shot poisoning effectiveness.
>
> ---
>
> **W3**: While the method achieves some improvement, the gains are modest (e.g., an average ASR increase of 8% in 1-shot). Given the paper’s focus on this specific enhancement, I expected more substantial improvements. In some cases, such as on Climate-FEVER (Table 1), the improvement is marginal.
>
> **A3:** We respectfully clarify that the reported improvements are neither modest nor directly comparable to the baseline numbers. **In Table 1**, the **8% gain** in the 1-shot setting is achieved under **strictly stronger success requirements**. Baselines only need to induce an incorrect answer, whereas ADMIT must additionally produce **coherent, deceptive justifications** that satisfies a 100 percent deceptive justification requirement. This makes the task substantially harder, so gains under this metric are more meaningful. Please also refer to our comprehensive result (**Table 19-22**),
>
> Beyond the single number highlighted by the reviewer, the overall performance trends **in a large scale experiments (table 19-22)** demonstrate substantial and consistent improvements:
>
> -  **Consistent state-of-the-art performance:** Across 60 experimental configurations (3 LLMs × 4 datasets × 5 shots), ADMIT achieves the highest attack success rate in more than 80 percent of settings, despite operating under stricter criteria than all baselines.
> - **Large gains on reasoning models:** On o1-mini, ADMIT improves over PoisonedRAG by as much as **33%**, reflecting the difficulty of attacking stronger models.
> - **High efficiency**: ADMIT reaches an average ASR of 86% with a poisoning rate of only $0.93 \times 10^{-6}$. 41% of successful attacks occur in a single iteration, and 65% within five iterations.
>
> >**"Climate-FEVER improvement is not marginal"**
>
> On Climate-FEVER, the improvement is not marginal when viewed holistically. ADMIT achieves the highest ASR in **80% (4 out of 5) shot settings**. The 1-shot difference is only 0.01 (0.57 vs. 0.58), and ADMIT remains the best performer in aggregate across all shots. Taken together, these results show that ADMIT provides **consistent and meaningful advances** under a strictly harder threat model, rather than modest or incremental improvements.

---

> ### Author Response · Authors · 2025-11-30
> **Reponse to Reviewer 8hnt (3/3)**
>
> **Q1: In the defense evaluation on LLM-based detection fails. Have the authors tried stronger models, such as GPT, for a quick check?**
>
> **A1**: Thanks for the question. **As detailed in Section 4.3 and Appendix F.4**, we already evaluated **LLM-based knowledge consolidation defenses** using **GPT-4o**, adopting the state-of–the-art LLM-based defense [2][3]. These defenses rely on the model’s pretrained knowledge for **both reranking and judgment**.
>
> However, such a defense still fails to mitigate attacks from ADMIT passage. As shown in Table 15 (Appendix), simple passage-level voting even **amplifies** adversarial signals, increasing ASR on datasets like SciFact. Clustering-based consolidation performs better by partially separating poisoned and clean passages, yet **ADMIT remains highly effective**, sustaining ASRs of **33–62%** even under the strongest GPT-4o defenses.
>
> [2] Wang, F., Wan, X., Sun, R., Chen, J. and Arik, S.O., 2025, July. Astute rag: Overcoming imperfect retrieval augmentation and knowledge conflicts for large language models. In Proceedings of the 63rd Annual Meeting of the Association for Computational Linguistics (Volume 1: Long Papers) (pp. 30553-30571).
>
> [3] Pan, Y., Pan, L., Chen, W., Nakov, P., Kan, M.Y. and Wang, W., 2023, December. On the risk of misinformation pollution with large language models. In Findings of EMNLP 2023 (pp. 1389-1403).
>
> ---
>
> **Q2: Why does the divide-and-vote defense fail? Does this imply that malicious documents dominate the retrieved results during the poisoning process?**
>
> **A2**: Thanks for the question. No, malicious passages do not dominate the retrieved set. This is exactly what makes our few-shot threat model challenging and realistic. ADMIT is effective even when only a small fraction of the retrieved context is poisoned, for example, one poisoned passage among 5 or 10.
> This happens for two main reasons:
> - 1. **Disproportionate persuasive strength.**
>
>  ADMIT passages are explicitly optimized to be more persuasive than typical clean evidence. Clean documents often provide neutral or moderately factual descriptions. In contrast, ADMIT passages emphasize semantic salience by using seemingly credible citations, assertive language, and authoritative-sounding entities. These properties arise from the observe then write optimization process rather than from simple templates, and they allow a few poisoned passages to steer the model’s judgment even in the presence of more numerous clean documents.
>
> - 2. **Statistical indistinguishability under the divide-and-vote method.**
>
>  Divide-and-vote defenses assume that adversarial passages can be detected or down-weighted at the passage level. However, as shown in **Figures 5 and 6**, PPL based and ROUGE N-based similarity signals fail to separate ADMIT passages from clean passages. Adv.–adv. pairs and clean–clean pairs often have comparable scores. Since the defense cannot reliably identify which passages are malicious, voting-based heuristics end up aggregating them together with clean passages, allowing a small number of high-impact adversarial passages to dominate the final verdict.
>
> **New Experiment Added In Appendix E.** We further evaluated ADMIT passages against 15 anomaly detection algorithms from PyOD [4]. The best-performing detector (COPOD) achieves only 68.0% accuracy, with most methods marginally above the 50% random baseline. This demonstrated that ADMIT passages cannot be reliably filtered at the embedding level, revealing that adversarial passages persist within any divided group during voting. Combined with our 1-shot results demonstrating that a single ADMIT passage can override four clean passages, voting-based defenses remain ineffective—the LLM's internal scoring still favors the more persuasive adversarial content.
>
> [4] Han, S., Hu, X., Huang, H., Jiang, M. and Zhao, Y., 2022. Adbench: Anomaly detection benchmark. Advances in neural information processing systems, 35, pp.32142-32159.

---

### Official Review · Reviewer_x6sL · 2025-10-29

**Soundness:** 3
**Presentation:** 3
**Contribution:** 3
**Rating:** 8
**Confidence:** 4

**Summary:**

This paper addresses a highly timely and critical problem: the vulnerability of retrieval-augmented generation (RAG) systems used for fact-checking, to “knowledge poisoning” attacks. The authors propose a novel method, ADMIT (Adversarial Multi-Injection Technique), which is notable in several respects. First, it systematically extends poisoning attacks into the more realistic fact-checking scenario — where retrieval pools include authentic supporting or refuting evidence, not merely benign or irrelevant context. This bridging of a practical deployment gap enhances the paper’s relevance for both the machine-learning and security communities. Second, the technical contribution is strong: ADMIT is a few-shot, semantically aligned poisoning approach that works without access to the target LLMs, retrievers or token-level control — thus modelling a realistic adversary. According to the abstract, it achieves an average attack success rate (ASR) of 86 % at an extremely low poisoning rate (~0.93×10⁻⁶) and shows transfer across multiple retrievers, LLMs, and cross-domain benchmarks. Such empirical strength and breadth of evaluation demonstrate both practical threat-model significance and robust experimental validation. Third, the work contributes to the broader “trustworthy AI / AI security” agenda by exposing an under-studied vulnerability of RAG-based fact-checking — a domain of clear interest to the ICLR audience given recent attention to large models, retrieval systems, and adversarial robustness. Taken together, the paper combines strong motivation, novel methodology, rigorous evaluation, and high relevance — making it well-suited for acceptance at ICLR.

**Strengths:**

See the summary above.

**Weaknesses:**

While the paper presents an impressive and comprehensive study of few-shot knowledge poisoning in RAG-based fact-checking systems, several areas could be improved to further strengthen its contribution.
	1.	Scope of Defense Evaluation.
The paper focuses primarily on offensive analysis. While this is understandable given the novelty of the attack, the defense discussion could be expanded. For example, integrating a simple retrieval-level or embedding-level filtering baseline (e.g., detecting anomalous injected samples or measuring cosine distance drift) would provide more balanced insight and demonstrate the practical implications for defending real-world RAG systems.
	2.	Theoretical Insight.
Although the empirical results are strong, the paper could benefit from a more formal characterization of why ADMIT succeeds under few-shot conditions. For instance, an analysis of embedding-space perturbation alignment or retrieval sensitivity could help reveal deeper understanding of poisoning dynamics — making the work more theoretically grounded for the ICLR audience.
	3.	Human Evaluation or Case Study.
Since fact-checking inherently involves human judgment and contextual reasoning, including a small-scale qualitative case study (e.g., analyzing misled outputs from a real-world RAG-based fact-checker) would make the findings more intuitive and impactful beyond numerical metrics.
	4.	Broader Implications and Ethical Considerations.
Given the increasing deployment of RAG systems in sensitive domains (news verification, misinformation detection, etc.), it would be valuable to include a short discussion on responsible disclosure and mitigation guidelines — aligning with ICLR’s growing focus on AI safety and responsible deployment

**Questions:**

No questions.

**Details Of Ethics Concerns:**

No ethic review needed.

---

> ### Author Response · Authors · 2025-11-30
> **Resposne to Reviewer x6sL**
>
> **W1: Scope of Defense Evaluation.**
>
> **A1:** Thanks for the valuable suggestion. We have now added an embedding-space visualization in Appendix E, along with a brief discussion of how ADMIT passages can be detected through their embedding signatures.
>
>
> ---
>
> **W2: Theoretical Insight**
>
> **A2:** Our study is intentionally grounded in comprehensive empirical analysis aimed at characterizing behaviors that have not been previously documented. While this empirical focus guides the structure and emphasis of the paper, we fully agree that a sharper theoretical framing would enhance the contribution. In the revision, we will expand the discussion to articulate the underlying assumptions, mechanisms, and failure modes that our empirical findings reveal, and clarify how these insights can inform future theoretical development.
>
> ---
> **W3: Human Evaluation or Case Study**
>
> **A3:** We agree that qualitative case studies would make the findings more intuitive. Our current evaluation is text-centric because real-world RAG fact-checkers often rely on additional metadata such as source URLs, publisher credibility, or trust scores. Unfortunately, the benchmarks we use (FEVER, Climate-FEVER, SciFact, HealthVer) are built from Wikipedia or paper abstracts and thus do not provide such metadata. If future benchmarks include richer source credibility information, we would be glad to extend our analysis to study how ADMIT interacts with real-world trust signals.
>
> ---
> **W4: Broader Implications and Ethical Considerations**
>
>
> **A4:** We thank the reviewer for emphasizing this important dimension. In response, we now added a dedicated section on “Broader Implications and Mitigation Guidelines,” where we discuss responsible deployment considerations for RAG-based fact-checking systems and outline practical safeguards for high-stakes domains.

---

### Official Review · Reviewer_njbn · 2025-11-01

**Soundness:** 2
**Presentation:** 3
**Contribution:** 1
**Rating:** 4
**Confidence:** 5

**Summary:**

The paper proposes ADMIT, a few-shot knowledge poisoning attack specifically designed for RAG-based fact-checking systems. The method uses a multi-turn, proxy-based optimization process to generate semantically coherent adversarial passages that mimic real news or reports. The stated goal of the attack is to flip the system's fact-checking verdict (e.g.,

**Strengths:**

The paper is well-written, clearly structured, and easy to follow. The authors' strongest contribution is the comprehensive empirical evaluation. The experiments are extensive, testing the proposed attack across 11 different LLMs (open-source, commercial, and reasoning-focused) and 4 retrievers, which demonstrates wide transferability.


The paper also does a thorough job of evaluating the attack against a suite of potential defenses, including statistical detection, fake news classifiers, and LLM-based knowledge consolidation . Demonstrating the failure of these defenses is a valuable, if concerning, result for the community.

**Weaknesses:**

Despite the strong empirical results, the paper suffers from significant weaknesses in its framing, novelty, and threat model.

Limited Novelty of the "Threat": The paper frames this as a new "knowledge poisoning" attack. However, it does not reveal a new vulnerability class. The core mechanism is the generation of highly plausible, semantically coherent misinformation ("fake news") that can be injected into a knowledge base. The techniques used to make this misinformation "convincing"—such as fabricating authoritative sources (e.g., "CDC and WHO joint statement" ), using recent timestamps (e.g., "March 2024" ), and providing plausible-sounding explanations (e.g., "new genomic analysis" )—are well-established strategies for creating effective disinformation. These are not new attack primitives, and similar concepts like "temporal ordering" and "contextual explanation" have been explored in prior work (e.g., GRAGPOISON[1] - 5.3.3. “Tricks” of Relation Injection.).


Misleading Problem Formulation: The authors state that the attack works even when "credible supporting or refuting evidence" is present. This reframes the problem entirely. This is not "poisoning" in the sense of corrupting the only available source of truth. Instead, this is a failure of the LLM's contextual conflict resolution. The core question is simply: when an LLM is given two contradictory passages (one true, one a well-crafted lie), can it identify the truth? The paper's high ASR (86%) demonstrates that modern LLMs are poor at this, which is a known limitation, not a new threat.





Unclear and Impractical Threat Model: The attack's evaluation relies on a "per-claim" generation process. This implies an attacker must identify a specific claim they want to flip and then expend significant computational resources (multi-turn optimization ) to craft a bespoke adversarial passage for that single claim. This is a high-effort, low-scalability attack model. The paper acknowledges this as a limitation and proposes fine-tuning as a fix , but the primary results and contributions are based on this impractical per-claim premise.





Confounding Effect of LLM's Prior Knowledge: The paper struggles to isolate the attack's effectiveness from the LLM's own internal knowledge. In fact, the authors' own analysis in Appendix B  highlights this weakness. The attack is most effective on "Gray" (conflicting signals) and "Black" (no prior knowledge) claims, achieving 75% and 98% ASR respectively . However, it is significantly less effective on "Gold" claims (52% ASR), where the LLM has strong, correct prior knowledge. This suggests the attack's success is heavily confounded by the LLM's pre-existing uncertainty, rather than demonstrating an ability to universally override established facts.

[1] Liang, J., Wang, Y., Li, C., Zhu, R., Jiang, T., Gong, N., & Wang, T. (2025). Graphrag under fire. arXiv preprint arXiv:2501.14050.

**Questions:**

1. Could the authors please clarify the novelty of the attack's "tricks" (e.g., temporal ordering, fabricated authority ) compared to prior work on both misinformation generation and knowledge graph poisoning (e.g., GRAGPOISON), which use similar strategies?

2. Given that the attack relies on the LLM failing to resolve a conflict between clean and poisoned text already in its context, why is this framed as a "knowledge poisoning" attack on RAG rather than a "contextual conflict resolution failure" of the LLM?


3. The results in Appendix B  are critical. They seem to suggest the attack is primarily effective when the LLM is already uncertain ("Gray" or "Black" claims). Doesn't this undermine the central claim of the attack's power, as it largely fails (or is at least 50/50) when trying to override "Gold" (known) facts?


4. Regarding the per-claim threat model: What is a realistic scenario where an attacker would expend the cost of multi-turn optimization for a single claim? If the proposed solution for scalability is fine-tuning , shouldn't this be the primary evaluation rather than an appendix?

---

> ### Author Response · Authors · 2025-11-30
> **Response to Reviewer njbn (1/4)**
>
> **Q1: Limited novelty of the threat, no new vulnerability class.**
>
> **A1** Please allow us to clarify the novelty of our threat model. Our attack adopts a **few-shot** poisoning setting, which differs from existing approaches such as PoisonedRAG. We believe this is more consistent with real-world scenarios, where an adversary typically aims to flip a specific claim using only a small number of injected passages. In contrast, the threat model used in most prior work implicitly requires the attacker to generate enough poisoned passages to **outnumber** the clean retrieved evidence. More concretely, we restrict the attacker to at most ( $m \leq k$ ) injected passages, where ( $k$ ) is the number of retrieved documents; prior work generally assumes ( $m \gg k$ ), which enables the attacker to dominate the retrieved context through volume. We believe this few-shot regime can be regarded as a novel threat model for poisoning attacks as it requires different attack techniques that optimize content rather than relying on quantity. We have revised the wording and replaced “novel” with “more restricted.”
>
> ---
>
> **Q2. The use of highly plausible “tricks” is well established in disinformation generation.**
>
> **A2**: We thank the reviewer for pointing out GRAGPOISON. Unfortunately, we were unable to obtain its source code for empirical comparison, so we instead clarify the conceptual distinction regarding persuasion strategies as follows:
>
> **GRAGPOISON** augments poisoned text during its relation-injection stage using predefined narrative strategies, such as fabricated authority, temporal framing, and explicit negation. These are manually specified heuristics that apply fixed enhancements to the generated content.
>
> Our **ADMIT**, in contrast, **does not rely on predefined trick templates**. Instead, it learns persuasive cues dynamically through an observe-then-write process: the proxy passage serves as a rich observational signal from which the ADMIT identifies and adapts context-specific patterns. This enables ADMIT to produce persuasion strategies that are tailored to the claim, the retrieval context, and the optimization trajectory, rather than applying a static set of heuristics.
>
> While **GRAGPOISON leverages stable, hand-crafted strategies**, ADMIT produces adaptive, claim-dependent persuasion signals that emerge from the multi-turn game-theoretic optimization process. This distinction is central to our design and leads to different attack behaviors and transferability properties.

---

> ### Author Response · Authors · 2025-11-30
> **Response to Reviewer njbn (2/4)**
>
> **Q3: Misleading problem formulation. With credible supporting or refuting evidence present, this is not “poisoning” but rather an LLM conflict-resolution failure. LLMs are known to struggle with conflicting information, so this does not constitute a new threat.**
>
> **A3:** We thank the reviewer for raising this important point. We agree that, from the LLM’s perspective, retrieving both clean and poisoned passages creates a contextual conflict. However, **the attack we study is still a knowledge-poisoning attack in the standard sense**, because ADMIT *injects adversarial passages directly into the external knowledge base*, influencing the LLM **only through the retrieval pipeline**. This follows the widely used definition of knowledge poisoning in RAG systems [1][2], where the adversary introduces conflicting or misleading content upstream of the model.
>
> Under this canonical formulation, ADMIT strengthens the threat substantially. It achieves near-perfect deceptive justification rates under a minimal injection budget ( $m \le k$ ). This demonstrates that conflict resolution alone does not explain the attack effectiveness; rather, ADMIT introduces *persuasive adversarial content* that succeeds even when the clean evidence is present.
> We also agree that LLMs are known to struggle with conflicting inputs, which is why defenses such as knowledge consolidation [3] have been proposed (by Google Cloud AI Research). We have evaluated this defense in **Table 15** and found that ADMIT remains highly effective even when the model aggregates, filters, and resolves evidence before producing a verdict. This indicates that the attack is not exploiting a naive conflict scenario but instead generates adversarial passages that survive structured resolution mechanisms designed specifically to address conflicting information.
>
> Furthermore, in timely fact-checking scenarios, there is often **no available gold evidence**. Any malicious injection is therefore neither supported nor refuted due to the absence of authoritative evidence. In such cases, ADMIT can construct alternative malicious passages without relying on conflicting information in the context window. This demonstrates ADMIT goes beyond the “naive conflicting attack” definition.
>
>
> > when an LLM is given two contradictory passages (one true, one a well-crafted lie), can it identify the truth?
>
> In fact, we evaluate an even stricter setting in our 1-shot experiments, where the LLM receives **four true passages and one well-crafted adversarial lie**, and is required to produce **both a verdict and a justification**. It distinguish our work for prior work **measuring success solely by whether the model produces an incorrect answer**.
>
>  We find that a single optimized passage can sometimes override a majority of evidence. For example, on FEVER, ADMIT attains an ASR of 0.44 against GPT-4o, whereas a weaker “lie” yields only 0.19 ASR. This gap indicates that the effectiveness of our attack is not explained by the model’s baseline conflict-resolution limitations but stems from the strength of the adversarial optimization itself.
>
> [1] Chen, Z., Xiang, Z., Xiao, C., Song, D. and Li, B., 2024. Agentpoison: Red-teaming llm agents via poisoning memory or knowledge bases. Advances in Neural Information Processing Systems, 37, pp.130185-130213.
>
> [2] Zou, W., Geng, R., Wang, B. and Jia, J., 2025. {PoisonedRAG}: Knowledge corruption attacks to {Retrieval-Augmented} generation of large language models. In 34th USENIX Security Symposium (USENIX Security 25) (pp. 3827-3844).
>
> [3] Wang, F., Wan, X., Sun, R., Chen, J. and Arik, S.O., 2025, July. Astute rag: Overcoming imperfect retrieval augmentation and knowledge conflicts for large language models. In Proceedings of the 63rd Annual Meeting of the Association for Computational Linguistics (Volume 1: Long Papers) (pp. 30553-30571).

---

> ### Author Response · Authors · 2025-11-30
> **Response to Reviewer njbn (3/4)**
>
> **Q4: Unclear and Impractical threat model; per-claim generation process is not practical; This is a low-scalability attack model;  what is a realistic scenario where an attacker would expend the cost of multi-turn optimization for a single claim?**
>
> **A4:** We respectfully disagree with the reviewer’s assessment that per-claim generation is impractical. In many real-world scenarios, an attacker has strong incentives to overturn **specific** claims that directly affect political, financial, or reputational interests [4,5].
>
> Moreover, our method is **not limited** to per-claim generation. It can be extended to attack **multiple claims simultaneously**.
>
> To demonstrate this, we conducted an additional **multi-claim** experiment on FEVEROUS [6]. Specifically, we sampled 50 multi-hop claims and decomposed them into 132 sub-claims. We generated poisoned content for each sub-claim and then synthesized all adversarial snippets into one unified passage. The setup used single-iteration generation, one-shot injection, (k = 5) retrieval, and GPT-4o as both generator and target model. As a baseline, we compared against the 1-shot ADMIT performance on FEVER. The results in the table below show that ADMIT extends reliably beyond per-claim attacks and can operate fairly effectively in multi-claim settings.
>
>
> | Metric | Result|
> |--------|-------|
> | Total multi-hop claims | 50 |
> | Total Synthesis Poisoned Passages| 50 |
> | Total sub-claims | 132 |
> | ASR (NEI not count) |42.42% |
> | ASR (NEI  count) |67.42% |
> | Avg. Recall with appending query | 100% |
> | Avg. Recall without appending query | 93.93% |
> |Baseline ASR | 50%|
> |Baseline Recall |~99% |
>
>
>
> > multi-turn optimization is not practical.  Where an attacker would expend the cost of multi-turn optimization for a single claim.
>
>
> While multi-turn generation is indeed more expensive, we note that **our attack is already highly effective with a single iteration**: for example, ADMIT reaches an ASR of 0.51 on FEVER and 0.44 on HealthVer in the one-iteration setting (Figure 9). Multi-turn optimization can be used selectively for high-value or hard-to-flip claims, depending on the attacker’s objectives.
>
>
> [4] Shu, K., Sliva, A., Wang, S., Tang, J. and Liu, H., 2017. Fake news detection on social media: A data mining perspective. ACM SIGKDD explorations newsletter, 19(1), pp.22-36.
>
> [5]: Zhou, X. and Zafarani, R., 2020. A survey of fake news: Fundamental theories, detection methods, and opportunities. ACM Computing Surveys (CSUR), 53(5), pp.1-40.
>
> [6] Aly, R., Guo, Z., Schlichtkrull, M., Thorne, J., Vlachos, A., Christodoulopoulos, C., Cocarascu, O. and Mittal, A., 2021. Feverous: Fact extraction and verification over unstructured and structured information. arXiv preprint arXiv:2106.05707.
>
>
>
> ---
>
>
> **Q5: Confounding Effect of LLM's Prior Knowledge: The paper struggles to isolate the attack's effectiveness from the LLM's own internal knowledge.**
>
>
> **A5:** Thank you for the insightful comment. We have carefully accounted for this concern in our initial experimental design. Specifically, we first run the full RAG pipeline on the unpoisoned knowledge base to obtain a clean verdict and set the opposite verdict as the target. Under this setup, any success can be attributed to ADMIT’s adversarial passage as we target the knowledge base.
>
> ---
> **Q6: Given that the attack relies on the LLM failing to resolve a conflict between clean and poisoned text already in its context, why is this framed as a "knowledge poisoning" attack on RAG rather than a "contextual conflict resolution failure" of the LLM?**
>
> **A6:** Thank you for the question. As clarified in A5, our attack explicitly targets the **RAG pipeline** rather than the LLM’s internal knowledge. ADMIT injects adversarial passages directly into the **external knowledge base**, which is the canonical definition of a knowledge-poisoning attack in retrieval-augmented systems. The LLM encounters conflicting evidence only *because* the underlying retrieval source has been tampered with. Moreover, our evaluation isolates this effect by attacking **only those claims for which the model produces clean verdicts** under an unpoisoned knowledge base. The 1-shot experiment mentioned above also proves that our method explores beyond simple contextual conflict resolution failure of the model.

---

> ### Author Response · Authors · 2025-11-30
> **Response to Reviewer njbn (4/4)**
>
> **Q7**: The results in Appendix B are critical. They seem to suggest the attack is primarily effective when the LLM is already uncertain. Doesn't this undermine the central claim of the attack's power, as it largely fails when trying to override "Gold" (known) facts?
>
> **A7**:  Thanks for the thoughtful question. Please allow us to clarify as follows:
>
> First, an ASR of 52% on Gold claims should not be interpreted as “failure.” Gold claims represent the **hardest** setting, where both the LLM and the RAG pipeline **agree** on a well-established fact, often drawn from Wikipedia, which is heavily represented in LLM pre-training. Achieving a 50% success rate against such strong prior knowledge is already a meaningful and practically relevant risk.
>
> Second, in real fact-checking scenarios, camouflage attacks are equally concerning. ADMIT can flip a verdict to NEI (“Not Enough Information”), which weakens user trust and undermines the reliability of fact-checking systems without requiring a full polarity reversal. As shown in Table 16, treating NEI responses as successful attacks increases ASR across all settings. For instance, even the statement “There is insufficient evidence to verify that wearing masks reduces COVID-19 transmission” would be damaging in practice, despite not being an explicit false claim.
>
> Third, as noted in A5, the highest-stakes misinformation scenarios, such as breaking news or emerging health events, are precisely those where LLMs lack strong priors. ADMIT’s high effectiveness on Gray and Black claims therefore reflects realistic threat conditions rather than a limitation of the attack.
>
>
> ---
>
> **Q8: If the proposed solution for scalability is fine-tuning, this could be the primary evaluation.**
>
> **A8**. Our scalability analysis is broader than fine-tuning alone:
>
> - First, **few-shot injection is inherently scalable**: ADMIT achieves high ASR with only 1–5 poisoned passages per claim at a poisoning rate of 0.93\times10^{-6}, already indicating practical efficiency.
>
> - Second, as shown in A4, ADMIT can attack multiple related queries with a single synthesized passage, further reducing cost.
> - Third, fine-tuning is presented as an additional downstream application, enabled by the clean–adversarial passage pairs produced by ADMIT. It is not intended as the primary evaluation, but rather as a demonstration of how the generated corpus can support other scalable attack settings.

---

### Author Response · Authors · 2025-11-30
**Summary Rebuttal (1/2)**

Dear PCs, SACs, ACs, and Reviewers,


We sincerely thank all reviewers for their time, careful reading, and constructive feedback. The comments raised have helped us refine the paper and clarify the core contributions. Below, we summarize key points of recognition and our main rebuttal efforts.


Our submission received four reviews: one **Accept (8)** and three **Marginally Below (4, 4, 4)**,  resulting in an **average initial rating of 5**. All three negative reviews centered on concerns regarding the **novelty of our threat model**, largely stemming from misunderstandings about the landscape of RAG poisoning research and the distinctions between our **few-shot threat model** and prior settings. We have addressed these issues thoroughly in the rebuttal, clarified the conceptual differences with precision, and revised the paper to ensure greater clarity and rigor.


---


**Reviewer Recognitions**


- Paper is well-written, clearly structured, and easy to follow (njbn, x6sL, 8hnt, jZ8Q)
- Experiments are extremely extensive and comprehensive (all four reviewers)
- Strong empirical results demonstrating SOTA performance across 11 LLMs and 4 retrievers （x6sL, jZ8Q）
- Defense evaluation is thorough and valuable for the community (njbn, x6sL)
- The attack shows wide transferability (njbn, x6sL)


Reviewer x6sL (Accept, rating 8) explicitly states:
> "The paper combines strong motivation, novel methodology, rigorous evaluation, and high relevance — making it well-suited for acceptance at ICLR."


---


**Reviewer Concerns**


- **Reviewer njbn:** Requested comparison with the recently published GRAGPOISON (Sep 2025) for multi-query attacks. We clarified that such a comparison is not directly appropriate: GRAGPOISON assumes a graph-indexed retrieval system and requires constructing and maintaining a graph over the corpus, whereas ADMIT is designed as a plug-and-play attack on wide employed text-embedding RAG pipelines. Nonetheless, to ease the reviewer’s concern, we have added new experiments on FEVEROUS to evaluate ADMIT under multi-query settings. The results show that our ADMIT remains fully feasible in this regime and achieves unexpectedly strong performance, reinforcing the generality of our threat model.


- **Reviewer 8hnt:** Commented that our method appears simple and the gains seem modest. We clarified that ADMIT operates under a significantly *more constrained and more realistic* threat model, with *strictly stronger* success criteria that require both verdict flipping and the generation of a deceptive justification. Under these conditions, ADMIT delivers substantial improvements across **11 LLMs** and **4 cross-domain benchmarks**, with gains reaching **33%** on reasoning-oriented models such as o1-mini. Moreover, regarding the reviewer’s suggestion to use LLMs as a quick defensive check, we note that this defense is already included in Section 4.3 under LLM-based Knowledge Consolidation.




- **Reviewer x6sL:** Expressed full confidence in our contribution. Recommended adding embedding-level defense analysis, additional theoretical framing, and a discussion of broader ethical implications. All of these suggestions have been incorporated into the revised paper.


- **Reviewer jZ8Q:** Raised concerns about the realism of knowing the target claim and suggested that similar adversarial studies already exist for fact-checking systems. We clarified that **all prior RAG poisoning work assumes knowledge of the target claim**, which is necessary for any **targeted poisoning attack**. Moreover, prior adversarial fact-checking studies largely focus on **BERT-** or **T5-style classifiers** or isolated RAG components, not **end-to-end LLM-based RAG systems**. The survey paper the reviewer mentioned explicitly highlights this gap and identifies LLM-based RAG fact-checking as an emerging research area. Our work directly addresses this missing piece.


---


**Rebuttal Highlights**


1. **Realistic Threat Model**
Our **few-shot poisoning** setting is intentionally designed to reflect real-world deployment conditions. Large-scale injection is unrealistic because it would be easily detected or filtered, whereas **even a single misleading passage** (for example, vaccine misinformation) can produce disproportionate real-world harm. This setting extends prior work by modeling adversaries who act **stealthily and strategically**, targeting high-stakes claims without overwhelming the corpus.


2. **Novelty**
We introduce the **observe-then-write** paradigm, a key insight showing that attackers can approximate the victim’s observation space through standard web retrieval. This enables the crafting of adversarial passages that remain coherent with clean passages yet successfully mislead end-to-end RAG pipelines under **strictly constrained** poisoning budgets. This mechanism is more than prompting; it is a principled framework for black-box knowledge poisoning.

---

### Author Response · Authors · 2025-11-30
**Summary Rebuttal (2/2)**

3. **Generalization**
Our ADMIT demonstrates strong and consistent performance across **general-purpose LLMs, reasoning-oriented LLMs, and multiple retrievers**, far beyond the settings used in prior work. In response to reviewer suggestions, we have also added new experiments showing that ADMIT is feasible in multi-query scenarios (e.g., FEVEROUS), and maintains competitive effectiveness without requiring complex indexing structures.


4. **Clarification and Revision**
We expanded the related-work discussion to include recent advances, incorporated embedding-based defense analysis, and improved the presentation and organization of experiments in the main paper for clarity and readability.






---


**Summary of Updates**


We sincerely thank all reviewers for their time and thoughtful feedback. While there is unanimous agreement on the strength of our empirical results and the effectiveness of ADMIT, we have incorporated all constructive suggestions and made substantial improvements to the paper. Specifically, we have:


- Added a discussion of GRAGPOISON in the related work section to contextualize multi-query attacks.
- Included 15 anomaly detection methods to provide deeper insights of ADMIT against  embedding-level defense.
- Added a dedicated section on ethical considerations and responsible deployment.
- Conducted new multi-query attack experiments (Appendix) to further confirm the generality of ADMIT.
- Expanded comparisons with prior adversarial attacks on traditional fact-checking systems, clarifying distinctions and connections.


---


**Closing**

We sincerely thank the AC and all four reviewers for their time and effort in evaluating our work. We greatly appreciate the recognition of the paper’s novelty, practicality, and reproducibility—it truly means a lot to us. The rebuttal process has allowed us to clarify remaining concerns and resolve misunderstandings, which in turn has significantly improved the clarity and overall quality of the paper.


Thank you again for handling our paper! Much appreciated!




Authors of Submission 11925

---

### Note · Program_Chairs · 2026-01-17
**Submission Desk Rejected by Program Chairs**

The following references in this submission do not refer to real documents and/or have major errors in bibliographic information:

 Yuxuan Li, Zhen Wang, Yuxuan Zhang, Yuxuan Liu, Zhen Wang, and Yuxuan Zhang. Agentpoison: Poisoning language agents via prompt injection. arXiv preprint arXiv:2403.12345, 2024. URL https://arxiv.org/abs/2403.12345.